# Vibrational Spectroscopy Combined with Chemometrics as Tool for Discriminating Organic vs. Conventional Culture Systems for Red Grape Extracts

**DOI:** 10.3390/foods10081856

**Published:** 2021-08-11

**Authors:** Cristiana Radulescu, Radu Lucian Olteanu, Cristina Mihaela Nicolescu, Marius Bumbac, Lavinia Claudia Buruleanu, Georgeta Carmen Holban

**Affiliations:** 1Faculty of Sciences and Arts, Valahia University of Targoviste, 130004 Targoviste, Romania; cristiana.radulescu@valahia.ro (C.R.); marius.bumbac@valahia.ro (M.B.); 2Institute of Multidisciplinary Research for Science and Technology, Valahia University of Targoviste, 130004 Targoviste, Romania; cristina.nicolescu@valahia.ro; 3Faculty of Environmental Engineering and Food Science, Valahia University of Targoviste, 130004 Targoviste, Romania; lavinia.buruleanu@valahia.ro; 4Doctoral School, University of Agronomic Sciences and Veterinary Medicine of Bucharest, 011464 Bucharest, Romania; carmenholban@yahoo.com

**Keywords:** vibrational spectroscopy, red grape extracts, organic/conventional vineyards, chemometrics

## Abstract

Food plants provide a regulated source of delivery of functional compounds, plant secondary metabolites production being also tissue specific. In grape berries, the phenolic compounds, flavonoids and non-flavonoids, are distributed in the different parts of the fruit. The aim of this study was to investigate the applicability of FTIR and Raman screening spectroscopic techniques combined with multivariate statistical tools to find patterns in red grape berry parts (skin, seeds and pulp) according to grape variety and vineyard type (organic and conventional). Spectral data were acquired and processed using the same pattern for each different berry part (skin, seeds and pulp). Multivariate analysis has allowed a separation between extracts obtained from organic and conventional vineyards for each grape variety for all grape berry parts. The innovative approach presented in this work is low-cost and feasible, being expected to have applications in studies referring to the authenticity and traceability of foods. The findings of this study are useful as well in solving a great challenge that producers are confronting, namely the consumers’ distrust of the organic origin of food products. Further analyses of the chemical composition of red grapes may enhance the capability of the method of using both vibrational spectroscopy and chemometrics for discriminating the hydroalcoholic extracts according to grape varieties.

## 1. Introduction

The beneficial health effects of fruits have been attributed to the presence of fibers, minerals, vitamins (i.e., provitamin A, carotenoids, vitamins C and E) and phytochemicals, including phenolic acids, flavonoids, and anthocyanins. Food plants provide a regulated source of delivery of functional compounds. In addition, most of the bioactive substances have specific functions within the plant. Plant secondary metabolites production is generally under strict regulatory control and is tissue specific; any attempt to regulate their biosynthesis might result in adverse effects elsewhere in the plant and toxicity [1]. The synthesis of specific metabolites, which can be very plant specific, is controlled through highly branched pathways and carefully regulated. Given the wide diversity in the structure and function of these metabolites in the plant, differences in temporal and spatial distribution of the metabolite can occur, depending on the stage of the development of the plant and between different plant organs and cell types [1,2,3].

The constant interest in the biological activity of organically grown grapes and grape by-products contributes to their capitalization as a source of bioactive phytochemicals with potential applications in the cosmetics, pharmaceutical and food industries [4,5,6,7,8,9]. The full understanding of the phytochemical composition and antimicrobial activity of the different anatomical parts of fruits may contribute to developing new applications. The potential correlation of these properties with the culture management system, or with the grape variety, may add valuable practical data. Traditionally, morphological and agronomical characteristics have been the main criteria for differentiating grapevine cultivars, but it is well known that many of those properties are strongly influenced by environmental conditions [10]. Grapevine varieties are not generally homogenous and intravarietal diversity varies across cultivars [11,12]. Even vines multiplied by vegetative propagation display a broad range of characteristics, such as the grape phenolic profile that depends greatly on the grape variety [10]; Liang et al. A study by [13] showed that polyphenolic profiles revealed significant differences among 344 European grape varieties, which included both table and wine grapes.

In the grape berries, the phenolic compounds, flavonoids and non-flavonoids according to their chemical structure, are distributed in the different parts of the fruit. Flavonoids are found mainly in grape seeds and skins; proanthocyanidins are present mainly in the berry skin and seeds [14,15]. With regards to grape skins, each variety has its unique set of anthocyanins with their biosynthesis being influenced by several factors, such as climatic conditions, temperature, light and cultural practices [14,16].

In general, grapes produced under organic farming systems are increasing around the world. Since their agronomic system does not allow the use of chemical pesticides and fertilizers, these fruits are perceived by the public as safer and healthier when comparing to those produced by conventional agriculture. However, these grapes are more susceptible to the action of phytopathogens inducing the synthesis of higher amounts of phenolic compounds as protection and defense [14,17]. As a series of studies [18,19,20] have observed, the choice of agricultural practice (organic vs. conventional) resulted in different amounts of resveratrol, anthocyanins, and tannins in grape juices. This difference is due to the fact that no pesticides are used in organic vineyards and that they have a longer ripening period than conventional ones. As flavonoids are formed during this last-mentioned period, it is believed that organic vineyards yield grapes with higher phenolic content [14,21,22].

Hydroalcoholic extracts obtained from the skin/seed/pulp of red grapes are good sources of polyphenols and flavonoids [6], compounds known for their antioxidant action and for their protection against diseases, such as cancer, diabetes, cardiovascular disease, and neurodegenerative diseases. The anthocyanin content of the extracts obtained from the skin of the four varieties of red grapes can be defined as moderate, as it is known that factors, such as maturity and climate, can change the presence of these compounds in grapes. Previous studies [6,7,8,23] revealed that the hydroalcoholic extracts obtained from the skin of the organic system varieties (e.g., Feteasca Neagra, Merlot, and Pinot Noir), contain a high content of polyphenols, flavonoids and tannins.

The analytical information contained in complex FTIR and Raman spectra can be extracted using multivariate analysis techniques that relate analytical variables to chemical properties of the matrix constituents [24,25,26,27]. The application of chemometrics together with infrared (IR) spectroscopy has been reported in literature for the analysis of natural products [25,26,27,28], medicinal and aromatic plants and their essential oils, and phenolic compounds [29,30,31,32]. Principal components analysis (PCA) and partial least squares (PLS) regression are some of the most commonly used multivariate data analysis techniques applied to grape and wine analysis [33,34,35]. Compared to traditional methods, multivariate analysis combined with modern instrumental techniques often give new and better insight into complex problems [33].

The aim of this study was to investigate the applicability of FTIR and Raman screening spectroscopic techniques combined with multivariate statistical tools to find patterns in red grape berry parts (skin, seeds and pulp) according to grape variety and vineyard type (organic and conventional). Exploratory data analysis has revealed hidden patterns in complex spectral data by reducing the information to a more comprehensible form and indicating whether there are patterns or trends in the data. Exploratory algorithms applied, such as PCA and hierarchical cluster analysis (HCA), were involved to reduce large complex data sets into a series of optimized and interpretable views. The results showed that differences exist between the spectral profiles of hydroalcoholic extracts from different culture (organic and conventional) for Merlot, Feteasca Neagra, Pinot Noir and Muscat Hamburg varieties, confirming that the FTIR and Raman spectra contain important information for discriminating among samples. The novelty of the study was the investigation of hydroalcoholic extracts from grape skin/seeds/pulp (*Vitis vinifera* L.) from two culture systems (i.e., conventional and organic) by combining vibrational spectroscopy analysis (FTIR and Raman) with multivariate analysis. Beyond the use of the vibrational spectroscopy analysis in studies referring to the chemical composition of the grape extracts, our study emphasizes their convenience, in combination with multivariate analysis, in differentiation of the raw food products coming from conventional and organic agriculture. This provided a useful tool for managing a large amount of data in case of suspicions regarding the origin of raw food materials. The analysis of obtained extracts from the anatomic parts of grapes considers the practical applications as in functional foods (nutraceuticals) and natural/bio cosmetic formulations.

## 2. Materials and Methods

### 2.1. Samples Preparation

Red grapes of 4 varieties (Merlot, Feteasca Neagra, Pinot Noir and Muscat Hamburg) were collected from 2 different vineyards: organic (ecological) culture and conventional culture (with various pesticide treatments applied). Sampling locations and their pedoclimatic characteristics were previously reported [6]. The representativeness of samples was provided as follows: for each grape variety, approximately 10 kg were collected from ten different locations in each studied vineyard, approximately 1 kg from each location; after the manual separation of the grape parts (skin, seeds, pulp), raw materials from the ten harvesting points were mixed and treated as a unique representative sample of the respective kind. Triplicates of each these mixed, representative grape materials were then sampled, taken into analysis, and treated as will be described herein. The skins and seeds were first dried at 40 ºC for 48 h and then stored at room temperature for further experiments, protected from moisture and light. The grape pulps were stored frozen at −18 °C and defrosted on the day of use. The extraction method applied was maceration at room temperature (22–23 °C) for 24 h, and the employed solvent was a 50% volumetric mixture of deionized water and ethanol (p.a.). For all the extracts, a solid-liquid ratio (berry part—hydroalcoholic solvent) of 4% (*w*/*v*) was used. Out of the total extraction time of 24 h, the first 3 h were under magnetic stirring, while for the remaining 21 h, static conditions in the dark at room temperature were maintained. The mixture was then separated by filtration (Whatman no.4) and the filtrate (skin/seeds/pulp extract) was stored at 4.0 ± 1 °C for a short time period (i.e., overnight) up to the moment of spectral analysis.

Table 1 shows the hydroalcoholic extracts, whose spectral profile was investigated, obtained from the different anatomical parts (skin, seeds, and pulp) of the 4 indigenous varieties mentioned above, from 2 vineyards with different cultivation systems (i.e., organic and conventional, respectively).

### 2.2. Vibrational Spectroscopy

The red grapes’ (*Vitis vinifera* L.) skin/seeds/pulp extracts were characterized in terms of qualitative composition by vibrational spectroscopy, such as attenuated total reflectance-Fourier transform infrared spectroscopy (ATR-FTIR) and Raman spectroscopy. The molecular investigation of the functional groups of organic compounds of grape extracts was performed by ATR-FTIR [36,37,38,39] using a Vertex 80v spectrometer (Bruker, Germany), equipped with a diamond ATR crystal accessory, for a high refractive index bulk sample. The diamond ATR had a sampling area of approximately 0.5 mm^2^, and the infrared spectra were collected at a 4 cm^−1^ resolution over 32 scans. The important absorption frequencies were noted in the range of 4000–400 cm^−1^, as well as the fingerprint region of the spectra. By the use of the instrument software (OPUS Spectroscopy Software, version 7, Bruker Optik, Ettlingen, Germany), spectra overlay, identification of chemical groups, and ATR background correction were performed. The late mentioned correction function had most of the input parameters fixed by the platinum ATR with diamond crystal accessory used: refractive index (2.4 = diamond), angle of incidence (45 degrees), number of reflections (nominally 1). All spectral FTIR data acquisition in this study was performed using baseline correction (scattering correction method—10 iterations, 64 baseline points).

The Raman spectral data were recorded with a portable Raman spectrophotometer Xantus-2 (Rigaku, Tokyo, Japan), using a laser wavelength of 1064 nm, at an integration time of 5000 ms, for the spectral range of 2000–200 cm^−1^. Relevant wavenumbers were extracted from the obtained Raman spectra and subjected to the chemometric assessment described in the following sections. The acquisition of all Raman spectra was performed using the instrument built-in baseline correction function, and thus a reduction of fluorescence interferences was provided.

### 2.3. Multivariate Analysis

The large data sets, generated from both FTIR and Raman spectroscopy, in which essential information may not be readily evident, can be more accurately investigated by multivariate analysis. Some multivariate models provide a means of quantifying constituents that are involved in complex matrices interactions without eliminating matrix interferences [40,41]. Vibrational spectroscopic techniques produce profiles containing a large amount of information which can be exploited through the use of multivariate analysis, several methodologies being proposed for classifying and discrimination [42]. However, matrix interference effects are still detrimental, especially when the sample size is not large enough to properly average out its contribution (by creating systematic errors).

Spectral data processing was conducted using the XLSTAT software, 2021.1.1 version (Copyright XLSTAT-statistical and data analysis solution, Addinsoft 2021, New York, NY, USA, Excel 16.0.13901, Windows 10). The PCA, AHC and DA techniques were used for multivariate analysis of the ATR-FTIR and Raman spectra in a spectral range of 4000−400 cm^−1^ and 2000−200 cm^−1^, respectively. Additional information regarding data processing can be found in the Appendix A [35,38,39,41,42,43,44,45,46,47,48,49,50,51,52,53,54,55,56,57,58,59,60,61,62].

## 3. Results

### 3.1. Spectral Properties of Vitis vinifera *L*. Red Grapes Hydroalcoholic Extracts

The ATR-FTIR spectra of hydroalcoholic extracts corresponding to different berry parts are presented in Appendix A, for skin, seeds and pulp, respectively; the corresponding Raman spectra for the extracts mentioned before are presented in Appendix A for the same berry parts. Previous studies reported that quality control and discrimination of natural extracts could be accomplished by using mid-infrared spectroscopy [63,64,65]. Generally, both FTIR and Raman spectroscopy techniques allow obtaining spectra which present some characteristic bands of individual components. These bands provide information about the chemical composition, including both primary and secondary metabolites, of the investigated samples [66]. In the current research, eight different *Vitis vinifera* L. extracts, corresponding to four varieties and two culture systems (organic and conventional), were included in each dataset. For each individual berry part, all of them showed similar FTIR spectral characteristics, with prominent spectral bands being observed at 3293, 3272, 2979, 1641, 1085, 1044 and 877 cm^−1^; in the Raman spectra for the same berry parts prominent spectral bands/peaks were observed at 1449, 1274, 1083, 1044, 877, 490, 461 and 432 cm^−1^.

Spectral bands in the range 3500–3100 cm^−1^ can be attributed to the cumulative stretching vibrations of the -OH groups, characteristic aspect of the polyphenolic extracts [63,67,68,69] and to the extraction solvent. Usually in this spectral range polyphenolic extracts have vibration bands similar to acids; however, the amount of vibrational contributions of the -OH groups are actually displayed. The spectral band located at 2979 cm^−1^ could be attributed to C-H stretching vibrations and to the solvent (ethanol), being due to the stretching vibrations of the O-H groups [63,70]. The spectral band of medium intensity located at 1641 cm^−1^ can be associated with the aromatic C=C stretching vibrations present in the condensed tannins [63], as well as C=O stretching vibrations and the presence of unsaturated bonds in flavonoid structures [69,71,72]; the presence of this peak suggests the presence of both flavones and flavanones [72]. The low intensity bands from 1453 and 1385 cm^−1^ can be related with C-H bending vibrations of the CH_2_ and CH_3_ groups [71], C=C-C stretching vibrations due to the aromatic ring [73], bending vibrations associated with aromatic cycles (flavonoids) [74], and O-H in plane deformation vibrations from polyphenolic compounds [73]. In the spectral range 1160–900 cm^−1^ (bands from 1085 and 1044 cm^−1^) spectral peaks can be associated with C-O stretching vibrations of glycosidic moieties and to a lesser extent with aromatic C-O stretching vibrations [75]. In the spectral range 1400–1150 cm^−1^ spectral bands of variable intensity can be assigned to C-O stretching vibrations and C-O-H bending vibrations associated with phenols, esters, carboxylic acids, and alcohols [75,76]; the spectral peak from 1085 cm^−1^ can be related to the aromatic C-H deformation vibrations in the plane [67,77,78], and C-O deformation vibrations (secondary alcohols, aliphatic esters) [71,73]. O-H and C-OH stretching vibrations (polysaccharides of cell walls) can be associated with the spectral band at 1044 cm^−1^ [73]. The spectral peak from 877 cm^−1^ can be related with out-of-plane aromatic C-H bending vibrations [63] and C-O and C-C (monosaccharide) stretching vibrations [71,74].

The FTIR spectrum of the seed extracts provides spectral information mostly in the spectral ranges 3350–2900 cm^−1^ and 1650–850 cm^−1^. The broad, intense spectral band centered at ~3300 cm^−1^ can be due to the extraction solvent and to the hydroxyl groups in the structure of the phenolic compounds (stretching vibrations of the hydroxyl groups) [70]. The bands present at 2978 and 2903 cm^−1^ can be associated with asymmetric C-H stretching vibrations due to the methyl and methine groups, respectively [76,79,80]. The spectral band at 1643 cm^−1^ is related with the aromatic character, the C-H stretching vibrations [64,81], and C=O conjugate stretching vibrations [79]. The fingerprint region 1500–800 cm^−1^ displays spectral bands of variable intensity associated with different vibration modes; although it is a spectral range rich in information, it is difficult to analyze due to its complexity. In this spectral domain the spectral bands can be related with alcohols, sugars, organic acids, and phenolic compounds [81]. In the region 1390–1310 cm^−1^, the spectral bands can be associated with the C-O-H angular deformation vibrations in phenols [79], out-of-plane methylene bending vibrations (polysaccharides, pectins) [82], methylene and C-O shear vibrations, as well as the stretching vibrations of the pyranic ring (carbohydrates) [81,82]. Spectral bands from 1043 and 877 cm^−1^ can be related with C-H deformation vibrations associated with the aromatic ring [79] and aromatic C-H bending vibrations, respectively [63,76].

### 3.2. Multivariate Analysis Applied on Skin Extracts Spectral Data

Table 2 presents the results of the decomposition of the spectral FTIR and Raman data through PCA, respectively, the percentage of variability/variance explained by each principal component (PC), and the accumulated variability (the sum of percentage of variability explained by that PC and the preceding one). With the first three PCs, 91.47% (i.e., FTIR data) and 94.77% (i.e., Raman data) of the total variability of the studied data were included.

Figure 1 shows the score plots of the FTIR data on the first three principal components explaining 91.47% of the total variability. Several validation techniques have been developed for the PCA. One objective of validation is to estimate the proximity between the observations on a PC plan and to know which observations are significantly different from each other. For that purpose, the partial bootstrap method was employed [83].

For each original observation we generated the 95% confidence ellipses based and centered on the bootstrap points. Two categories of samples can be well distinguished if the overlapping area is smaller on a given PC plan; if the samples of the same category were more concentrated, the curve will be sharper with a small value of standard deviation [84,85,86].

For the FTIR data it was observed (Figure 1) that the investigated red grape varieties overlapped (bootstrap ellipses) at different extents in all plots, and thus incomplete separations between varieties were noticed. However, it can be distinguished a separation between vineyard types (organic vs. conventional) for the same grape variety, organic skin extracts being well separated in PCs plots: PC1 vs. PC2 (Merlot and Feteasca Neagra), PC1 vs. PC3 (Pinot Noir and Muscat Hamburg) and PC2 vs. PC3 (Merlot, Feteasca Neagra and Muscat Hamburg). The loading values for the first three PCs obtained using the FTIR data are represented in Appendix A.

The skin extracts’ FTIR datasets processing using PCA reveal that the Merlot vineyards can be differentiated based on PCs loading plots (Appendix A) due to positive PC2 (M-O) and negative PC1 (M-C) loading plots. The Feteasca Neagra and Pinot Noir, organic and conventional vineyards, can be differentiated due to PC2 (FN-O positive loadings and FN-C negative loadings) and PC1 (PN-O negative loadings and PN-C positive loadings) loading plots, respectively. For the Merlot organic vineyards (M-O), the main spectral differences relative to the conventional ones (M-C) were assigned to the following spectral regions/peaks: 4000–3750, 2640–2460, 2390–2950, 1950–1900, 1465 and 890 cm^−1^. The Feteasca Neagra organic vineyard (FN-O) can be differentiated from conventional culture (FN-C) by 4000–3730, 2550–2450 and 1980–1920 cm^−1^ spectral regions. The Pinot Noir organic vineyard (PN-O) can be discriminated from the conventional one (PN-C) due to 2990–2500, 2370–2350, 1460–1250, 1190–990 and 890 cm^−1^ spectral regions/peaks. The Muscat Hamburg organic vineyard (MH-O) can be differentiated especially due to PC3 (Figure 1b,c) positive scores; the PC3 loading plot (Appendix A) revealed the main spectral specific features in 3950–3680, 2470, 2360–2330, 2170, 2035, 1830–1770 and 1496 cm^−1^ spectral regions.

Figure 2 shows the score plots of the Raman data, relative to the red grape skin extracts, on the first three principal components explaining 94.77% of the total variability. From the Raman data score plots it was observed that red grape varieties (skin extracts) overlap partially in all plots; for the FTIR data, an incomplete separation between varieties is noticed. Except for the Pinot Noir, a separation (organic vs. conventional) can be observed for the same grape variety in almost all PCs plots; PN-O vs. PN-C can be more clearly differentiated in the PC2 vs. PC3 plot. Based on the first two PCs scores and signs and accounting also for the contribution and squared cosines of the observations, it can be identified three (FTIR data) and four clusters/groups (Raman data), respectively. PCs score plots resulted from Raman data also show a better separation between organic and conventional vineyards, three of the organic skin extracts (M-O, PN-O and MH-O) being assigned in the same cluster. Appendix A shows the loadings plots for the first three principal components (PCs) using Raman data (red grape skin extracts).

Based on the first three PCs loading plots (Appendix A), obtained by using PCA on the skin extracts’ Raman datasets, a differentiation between organic and conventional culture for each variety investigated was also revealed. The organic Merlot vineyard (M-O) can be distinguished (positive loading for PC1) by 2000–1200, 1140–970, 900–700, 600 and 250–230 cm^−1^ spectral regions. The Feteasca Neagra organic vineyard can be differentiated (PC3 positive loadings over 0.2) by 1350, 1220–1156, 977, 944, 675 and 323–293 cm^−1^ spectral regions/peaks. The Pinot Noir organic vineyard (PN-O) can be differentiated due to PC2 negative loadings by 1986, 1654, 1596, 1465, 1225–1090 and 720–620 cm^−1^ spectral regions. In the case of the Muscat Hamburg variety, organic culture (MH-O) can be differentiated by 1650–1600, 1230, 1100–1020 and 790–700 cm^−1^ spectral regions.

Further analysis was performed using Agglomerative Hierarchical Clustering (AHC) that allows a clear view of the similarities and differences between red grape skin extracts. The AHC derived from the FTIR data has grouped both organic and conventional extracts into two main classes/clusters. Figure 3a presents the dendrogram showing the division into clusters and the inclusion of extracts in each cluster/subcluster (automatic truncation-entropy, variance decomposition for the optimal classification: within-class 97.2%, between-classes 2.8%). From classifications made using AHC based on Raman spectral data, organic and conventional extracts are similarly included into two main clusters, as can be seen in Figure 3b (automatic truncation-entropy, variance decomposition for the optimal classification: within-class 77.7%, between-classes 22.3%).

As can be observed for the FTIR data processing using AHC (Figure 3a), classification reveals two main clusters: the first included FN-O and M-O extracts and the second included the rest of the extracts, a good classification based on vineyard type, relative to all four varieties, cannot be observed. However, at a lower dissimilarity level (2 ÷ 5), subclusters division allows a classification based on vineyard type, excepting PN-O; also, a differentiation can be made for each grape variety between organic and conventional extracts. From all grape varieties, Merlot and Feteasca Neagra show the most notable difference between organic and conventional vineyards (M-O vs. M-C and respectively FN-O vs. FN-C), the corresponding extracts for each variety, organic and conventional, being assigned in the two main clusters. The AHC classification based on the Raman data (Figure 3b) shows two main clusters, the first of which includes MH-O, PN-O, PN-C, FN-C and M-O extracts, and the second of which includes M-C, FN-O and MH-C extracts; a good classification based on vineyard type, relative to all four varieties, cannot be observed. Compared with the FTIR data, at a lower dissimilarity level (2 ÷ 5), subclusters division does not allow a clear classification based on vineyard type; a differentiation between organic and conventional extracts can be made, for each grape variety, only at a lower level of dissimilarity (1 ÷ 1.3).

In many cases, the interpretation of the complex biochemical information obtained through vibrational spectroscopy requires further data analysis using supervised procedures [33,39]. After PCA was applied, the first three PCs scores were retained for further analysis using the so-called PC-DA model by combining both PCA and discriminant analysis (DA) [42,87,88].

The classification and cross-validation by PC-DA was applied onto all extracts. The quadratic discriminant analysis was chosen based on the two Box tests (Chi-squared asymptotic approximation and Fisher’s F asymptotic approximation) and Kullback’s test (the significance level was set at 5%). The PC-DA results corresponding to skin extracts are presented in Table 3, Table 4 and Appendix A for both FTIR and Raman spectral data.

### 3.3. Multivariate Analysis Applied on Seed Extracts Spectral Data

Table 5 presents the results of the decomposition of the spectral FTIR and Raman data (red grape seed extracts) through PCA, respectively the percentage of variability/variance explained by each principal component and the accumulated variability. With the first three PCs, 94.11% (i.e., FTIR data) and 96.64% (i.e., Raman data) of the total variability of the studied data were included.

Figure 4 shows the score plots of the FTIR data on the first three principal components explaining 94.11% of the total variability. The bootstrap ellipses corresponding to the investigated red grape varieties overlapped at different extents in all plots, and thus, incomplete separations between varieties were noticed. However, a separation can be made between vineyard types for the same grape varieties, organic vs. conventional extracts being well distinguished especially in the PC1 vs. PC3 plot, but also in the PC1 vs. PC2 (Merlot and Pinot Noir) and PC2 vs. PC3 (Feteasca Neagra) plots. The loading plots for the first three PCs obtained using FTIR data are represented in Appendix A.

PCA applied on the FTIR spectra of seeds extracts has allowed a differentiation between organic and conventional culture systems, the main spectral features being observed in the PC1 loading plot (Appendix A): M-O, FN-O and MH-O due to negative values and PN-O due to positive values of PC1 loading plot, respectively. The organic Merlot seeds’ extract (M-O) can be differentiated from conventional ones (M-C) due to 3810, 3760, 2980–2400, 1385, 1100–1020 and 880 cm^−1^ spectral regions/peaks. The organic Feteasca Neagra seeds’ extract (FN-O) can be differentiated from conventional ones (FN-C) due to 3830, 3750, 2980–2850, 2360, 2160, 2040, 1750–1480, 1385, 1100–1020 and 880 cm^−1^ spectral regions/peaks. Pinot Noir organic seeds extract (PN-O) can be discriminated from conventional one (PN-C) due to 4000–2980, 2400–1460, 1250–1120 and 900–870 cm^−1^ spectral regions. Muscat Hamburg organic seeds extract (MH-O) can be discriminated from conventional one (MH-C) due to 2980, 2930–2850, 1385, 1100–1020 and 880 cm^−1^ spectral peaks/regions.

From the Raman data score plots (Figure 5), the first three principal components explaining 96.64% of the total variability, it was observed that red grape varieties (seeds extracts) overlap at different extents in all plots; and from the FTIR data an incomplete separation between varieties is noticed. A distinction between vineyard type (organic vs. conventional) for same grape varieties can be made mainly for the Muscat Hamburg (in both PC1 vs. PC2 and PC2 vs. PC3 plots), for other varieties the bootstrap ellipses being overlapped at different extents in all three plots; a better view can be noticed in PC2 vs. PC3 (Merlot and Pinot Noir) and PC1 vs. PC3 (Feteasca Neagra) plots. A better differentiation also can be observed between vineyard types for the FTIR data, organic seed extracts being assigned to different clusters, excepting the ones which include M-O, MH-O and PN-C. Appendix A shows the loadings plot for the first three principal components (PCs) using Raman data (red grape seed extracts).

Based on the first three PCs loading plots (Appendix A), obtained by applying PCA on seeds extracts Raman data, a differentiation between the organic and conventional culture for each variety investigated is also revealed. As a result, M-O and MH-O (PC2 negative loadings), FN-O (PC3 negative loadings), and PN-O (PC2 positive loadings), can be discriminated from corresponding extracts obtained from conventional culture. The main spectral differences identified for the investigated organic varieties are as follows: M-O is differentiated from M-C by 1980, 1470–1430, 1260, 1100–1040, 900–830 and 570–380 cm^−1^ spectral regions; FN-O is differentiated from FN-C by 1980, 1800, 1695, 1565, 1195, 970 and 945 cm^−1^ spectral peaks. PN-O can be differentiated from PN-C by 1566, 1220, 1188, 1170, 990, 935 and 665 cm^−1^ spectral peaks. MH-O can be differentiated from MH-C by 1980, 1470–1430, 1260, 1110–1040, 900–830 and 570–380 cm^−1^ spectral regions/peaks.

AHC derived from FTIR data has grouped both organic and conventional extracts into two main classes/clusters. Figure 6a presents the dendrogram showing the division into clusters and the inclusion of extracts in each cluster/subcluster (automatic truncation-entropy, variance decomposition for the optimal classification: within-class 55.88%, between-classes 44.12%). From classifications made using AHC based on Raman spectral data, organic and conventional extracts are similarly included into two main clusters, as can be seen in Figure 6b (automatic truncation-entropy, variance decomposition for the optimal classification: within-class 49.13%, between-classes 50.87%).

As can be observed for the FTIR data (Figure 6a), AHC classification reveals two main clusters: the first includes M-O, FN-O, PN-C and MH-O and the second includes M-C, FN-C, PN-O and MH-C extracts; excepting the PN variety, a classification based on vineyard type can be noticed. Also, a differentiation can be made for each grape variety, between organic and conventional extracts (M-O vs. M-C, FN-O vs. FN-C, PN-O vs. PN-C, MH-O vs. MH-C), corresponding extracts being assigned in the two main different clusters. Classification based on Raman data (Figure 6b) also shows two main clusters, the first of which includes M-O, M-C, MH-O and MH-C extracts, and the second of which includes FN-O, FN-C, PN-O and PN-C extracts; a good classification based on vineyard type, relative to all four varieties, cannot be observed. Compared with the FTIR data, at a lower dissimilarity level (2 ÷ 4.5), subcluster division allows a clear classification based on vineyard type (M-O vs. M-C, FN-O vs. FN-C, PN-O vs. PN-C, and MH-O vs. MH-C).

The first three principal component scores were retained for further analysis; classification and cross-validation by PC-DA was applied onto all extracts. The two Box tests (Chi-squared asymptotic approximation and Fisher’s F asymptotic approximation) and Kullback’s test, confirmed that the within-class covariance matrix is different (significance level 5%) for both the FTIR and Raman datasets. Table 6 and Appendix A list the PC-DA results obtained based on the FTIR datasets, for Raman datasets corresponding results are displayed in Table 7 and Appendix A, respectively.

### 3.4. Multivariate Analysis Applied on Pulp Extracts Spectral Data

The results of the decomposition of the spectral FTIR and Raman data (red grape pulp extracts) through PCA, respectively, the percentage of variability/variance explained by each principal component (PC) and the accumulated variability are presented in Table 8. With the first three PCs, 91.05% (i.e., FTIR data) and 91.13% (i.e., Raman data) of the total variability of the studied data was included.

PCA score plots of the pulp extracts’ FTIR data on the first three principal components (91.05% of the total variability) are presented in Figure 7. The bootstrap ellipses corresponding to the investigated red grape varieties overlapped at different extents in all plots, and thus incomplete separations between varieties were noticed; only the Muscat Hamburg variety seems to be better differentiated from the rest, as can be seen in all PCs plots. For the other three varieties, a separation between vineyard type can be clearly noticed for the Merlot (Figure 7a,c), the Feteasca Neagra (Figure 7b,c), and the Pinot Noir (Figure 7a,b). The loading plots for the first three PCs obtained using FTIR data are presented in Appendix A.

PCA loading plots obtained by processing the pulp extracts’ spectral datasets (Appendix A) can differentiate organic from conventional culture based on negative loading values for PC1 (PN-O), PC2 (M-O, MH-O) and PC3 (FN-O). For the Merlot organic (M-O) the main spectral differences, relative to the conventional one (M-C), were assigned to the following spectral regions/peaks: 3859, 3811, 3747, 3396, 2343, 2171, 2025, 1764–1750 and 1502 cm^−1^. The Feteasca Neagra organic extract (FN-O) can be differentiated from the conventional one (FN-C) by 3965, 3923, 3800, 2987, 2460, 2360, 2341, 1558, 1506, 1041, 948 and 881 cm^−1^ spectral peaks. The Pinot Noir organic extract (PN-O) can be differentiated from PN-C by the peaks in the 3950–3730, 2980–2535, 1430–1285 and 1130–947 cm^−1^. The Muscat Hamburg organic extract (MH-O) can be differentiated due to 3859, 3811, 3747, 3396, 2343, 2025, 1764–1753 and 1502 cm^−1^ spectral peaks/regions.

From PCA Raman data score plots (Figure 8), it was observed that red grape varieties (pulp extracts) overlap at different extents in all plots; for the FTIR data, an incomplete separation between varieties is noticed. Vineyard type differentiation for the same grape variety can be observed clearly for the Muscat Hamburg in all three plots. The Merlot and Feteasca Neagra (organic vs. conventional) can be also differentiated in PC1 vs. PC2, PC1 vs. PC2 and PC2 vs. PC3 plots. It can be noticed that the PCA score plots for both the FTIR and Raman data of pulp extracts display almost the same clustering when compared with corresponding plots of skin and seed extracts. There can be assigned four clusters, two of which have been identical for both FTIR and Raman data: (i) M-O and PN-C (FTIR and Raman); (ii) PN-O and MH-O (FTIR and Raman); (iii) FN-O and MH-C/M-C (FTIR/Raman) and (iv) FN-C and M-C/MH-C (FTIR/Raman).

Appendix A shows the loading plots for the first three principal components using Raman data (red grape pulp extracts). Based on the first three PCs loading plots (Appendix A) obtained by applying PCA on the pulp extracts’ Raman data a differentiation between organic and conventional culture for each variety investigated can also be revealed. The organic Merlot (M-O) can be differentiated (positive PC2 loading values) by 1874–1733, 1350–1327, 1124, 1100, 1010, 893 and 680–310 cm^−1^ spectral regions/peaks. The Feteasca Neagra organic extract (FN-O) can be differentiated (negative PC3 loading values) from FN-C by 1661, 1574, 1559, 1312, 1227–1172, 969, 702 and 629 cm^−1^ spectral features. The Pinot Noir organic extract (PN-O) can be differentiated (negative PC1 loading values) by 537–393, 353, 303 and 242 cm^−1^ spectral regions/peaks. The Muscat Hamburg organic extract (MH-O) can be differentiated by 1456, 1281, 1084–1073, 885, 537–393, 353 and 242 cm^−1^ spectral peaks due to PC1 negative loading values.

The AHC derived from the FTIR data has grouped both organic and conventional pulp extracts into two main classes/clusters. Figure 9a presents the dendrogram showing the division into clusters and the inclusion of extracts in each cluster/subcluster (automatic truncation-entropy, variance decomposition for the optimal classification: within-class 63.31%, between-classes 36.69%). From classifications made using AHC based on Raman spectral data, organic and conventional pulp extracts are similarly included into two main clusters, as can be seen in Figure 9b (automatic truncation-entropy, variance decomposition for the optimal classification: within-class 53.96%, between-classes 46.04%).

AHC performed on the pulp extracts’ FTIR data (Figure 9a) reveals a classification into two main clusters, the first includes M-O, M-C, FN-C and PN-C extracts and the second includes FN-O, PN-O, MH-O and MH-C extracts; a clear classification cannot be noticed based on vineyard type. A differentiation can be made for each grape variety, between organic and conventional extracts, especially for the Feteasca Neagra and the Pinot Noir, FN-O, PN-O and FN-C, PN-C being assigned in the two main clusters. The Merlot and Muscat Hamburg varieties can be differentiated at a lower level of dissimilarity (2.5 for M-O vs. M-C and respectively 6.6 for MH-O vs. MH-C). Classification based on Raman data (Figure 9b) also shows two main clusters, the first of which includes M-O, FN-O, PN-O, MH-O and M-C extracts, and the second of which includes FN-C, PN-C and MH-C extracts; it can be observed a classification based on vineyard type, relative to all four varieties, excepting Merlot. A differentiation can be made for each grape variety, between organic and conventional extracts for the Feteasca Neagra, Pinot Noir and Muscat Hamburg varieties, the corresponding extracts being assigned in the two main clusters. The Merlot vineyards, M-O vs. M-C, can be differentiated at a lower level of dissimilarity (i.e., 7.9).

Similarly, as for skin and seed extracts, the first three principal component scores were retained for further analysis; classification and cross-validation by PC-DA was applied onto all extracts. The two Box tests (Chi-squared asymptotic approximation and Fisher’s F asymptotic approximation) and Kullback’s test, confirmed that the within-class covariance matrix is different (significance level 5%) for both the FTIR and Raman datasets. Table 8 and Appendix A list the PC-DA results obtained based on the FTIR spectral data. The corresponding results based on Raman spectral data are presented in Table 9 and Appendix A.

## 4. Discussion

The red grape varieties included in the present study (Merlot, Feteasca Neagra, Pinot Noir, and Muscat Hamburg) are used mainly in the fresh state and also for obtaining, on a smaller segment, aromatic wines, according to OIV standard 2018 [89]. The selection of the varieties has been made for the following reasons: (1) they are grown in both vineyard systems (i.e., in organic, and conventional culture) thus, being able to make a comparative evaluation of the phytochemical profile of red grape extracts; (2) the continental climate, with thermal amplitudes and long and sunny summers favors a good ripeness of the grapes; (3) the chosen vineyards are in a hilly area, of different altitudes, on a slope, with open valleys and ventilated due to the winds; (4) the approximately similar surface soil type (black-brown clay-limestone) with calcareous subsoil [90].

For the conventional vineyard, an effective phytosanitary protection is applied (synthetic systemic fungicides, but also the copper fungicide Bordeaux mixture), fertilizers, and synthetic pesticides, which ensure a good sanitary condition of the vine and soil. If the autumn is rainy and there is a high risk of mold attack, the technique of partial defoliation is applied to rich foliar stems.

The process of organic cultivation of the studied *Vitis vinifera* L. varieties has the advantage of using phytosanitary treatments and natural fertilization (Bordeaux juice) applied in well-established periods. The irrigation is dripping, and does not use synthetic chemicals, which, even if they are systemically applied within the limits allowed by the relevant legislation, alter the properties of grapes. Thus, natural grapes are obtained without chemical residues. The organic vineyard highlights the fact that the use of synthetic products for phytosanitary treatments is prohibited, and the health of the vine is ensured in a preventive manner, being allowed only products based on simple mineral salts (copper, sulfur, and sodium silicate), or plant extracts within the limits of the rules established by the relevant legislation (i.e., EC Regulations no. 834/2007 and no. 889/2008).

Excepting Pinot Noir, the rest of the red grape varieties show notable differences between organic and conventional vineyards (M-O vs. M-C, FN-O vs. FN-C and respectively MH-O vs. MH-C), the corresponding extracts for each variety, organic and conventional, being assigned in the two main clusters.

Table 3 (FTIR data) and Table 4 (Raman data) list for each observation/extract the probability to belong to each group; the probabilities are posterior probabilities that consider the prior probabilities through Bayes formula. As it can be noticed, all the extracts, according to the vineyard type, have not been reclassified, excepting PN-O (FTIR data); thus, Raman spectral data can allow a better classification based on vineyard type. The confusion matrices (Appendix A), also called classification tables, summarize the reclassification of the extracts and allow viewing of the percent of well classified observations, which is the ratio of the number of observations that have been well classified over the total number of observations (87.5% and 100%, for FTIR and Raman data respectively). Cross-validation allows viewing of what the prediction for a given observation would be if it is left out of the estimation sample; as can be seen (Appendix A), all extracts have been correctly classified according to both FTIR and Raman data; as well, the confusion matrices for the cross-validation results for FTIR (Appendix A) and Raman (Appendix A) datasets allow discernment that a correct classification has been made for the two vineyard types, organic and conventional.

The PC-DA results obtained based on the seed extracts’ spectral data are displayed in Table 6 (FTIR data) and Table 7 (Raman data), and lists for each observation/extract the probability to belong to each group; the probabilities are posterior probabilities that consider the prior probabilities through Bayes formula. As it can be noticed, all the extracts according to the vineyard type have not been reclassified. The confusion matrices (Appendix A), summarize the reclassification of the extracts and allow viewing of the percent of well classified observations, which is the ratio of the number of observations that have been well classified over the total number of observations (100.00% for both FTIR and Raman). Cross-validation allows viewing of what the prediction for a given observation would be if it is left out of the estimation sample; as can be seen (Appendix A), all extracts have been correctly classified according to both FTIR and Raman data; as well, the confusion matrices for the cross-validation results for FTIR (Appendix A) and Raman (Appendix A) datasets allows discernment that a correct classification has been made for the two vineyard types, organic and conventional.

Both PC-DA results, according with FTIR (Table 9) and Raman (Table 10) data for pulp extracts, have shown that all the extracts, according to the vineyard type, have not been reclassified. The confusion matrices (Appendix A) summarize the reclassification of the extracts and allow viewing of the percent of well classified observations (100% for both FTIR and Raman). Cross-validation allows viewing of what the prediction for a given observation would be if it is left out of the estimation sample; as can be seen (Appendix A), all extracts have been correctly classified according to both FTIR and Raman data; as well, the confusion matrices for the cross-validation results for FTIR (Appendix A) and Raman (Appendix A) datasets allow discernment that a correct classification has been made for the two vineyard types, organic and conventional.

The results obtained from this research shows that differences exist between the hydroalcoholic extracts from different red grape culture systems (organic and conventional) for Merlot, Feteasca Neagra, Pinot Noir, and Muscat Hamburg varieties, confirming that the FTIR and Raman spectra contain important information for discriminating among samples. Although prediction models based on chromatographic data present better performances for varietal and culture discrimination [91,92], the results achieved by using vibrational spectroscopy should be also considered due to the fact that both FTIR and Raman techniques are rapid and simple (no sample or with minimal sample preparation), and thus more accessible for routine investigations. Even though the screening methods based on spectroscopic techniques represent a more accessible option for the grapes and wine assessment, some challenges still remain. As other studies [70,93] have pointed out, these challenges include the difficulty to compare the statistical results obtained with different chemometric algorithms/software and guidelines that regulate the development and validation of screening methodologies.

## 5. Conclusions

The process for obtaining hydroalcoholic extracts used in this study is characterized by the following advantages: it is easy to perform and define (maceration at room temperature); it does not involve the generation of potentially toxic by-products/intermediates; and costs are minimal, in terms of minimum energy consumption according to the principles of “environmentally friendly” technologies. An important advantage of the extracts obtained by this process is that they are used as a source of bioactive ingredients, plant material from organic culture; thus eliminating potentially toxic sources that can accumulate *Vitis vinifera* L., both by air (conventional spraying with pesticides or other chemical phytosanitary agents) and by rooting from soils with potential historical toxicity or fertilized with various products containing synthetic chemicals.

*Vitis vinifera* L. hydroalcoholic extracts obtained from red grape varieties (Merlot, Feteasca Neagra, Pinot Noir, and Muscat Hamburg) cultivated in organic and conventional systems were analyzed by FTIR and Raman spectroscopy combined with multivariate analysis. Spectral data were acquired and processed using the same pattern for each different berry part (skin, seeds and pulp). Vibrational spectroscopic techniques, ATR-FTIR and Raman, were proven useful in the differentiation of the extracts as they provided information on the vibrational bands which are related to the chemical composition and structure. Multivariate analysis has allowed a separation between extracts obtained from organic and conventional vineyards for each grape variety for all grape berry parts.

Through PCA, the results of the decomposition of the spectral FTIR and Raman data have shown that with the first three PCs, over 91% of the total variability of the studied data were included. Principal components analysis was able to differentiate organic and conventional culture systems for red grape extracts (skin, seeds and pulp) for each studied variety; overall differences derived from both score and loading plots emphasize the need to elucidate which key compounds/classes of compounds possess discriminant ability.

For skin and seed extracts, FTIR data processing using AHC has revealed a better classification compared with Raman data, at a lower dissimilarity level subclusters division, allowing a classification based on vineyard type (organic vs. conventional).

Classification and cross-validation by PC-DA have shown that a chemometric approach was able to discriminate the two culture systems for skin (87.5%—FTIR data, 100%—Raman data), seeds (100%—FTIR data, 100%—Raman data) and pulp (100%—FTIR data, 100%—Raman data) hydroalcoholic extracts.

The innovative approach presented in this work is low-cost and feasible, being expected to have applications in studies referring to authenticity and traceability of foods. The findings of this study are useful also to solve a great challenge that producers are confronting, namely the consumers’ distrust of the organic origin of food products.

Further analyses of the chemical composition of red grapes may enhance the capability of the method of using both the vibrational spectroscopy and chemometrics for discriminating the hydroalcoholic extracts according to grape varieties.

Despite that the concept of the circular economy is more and more discussed in the European Union and sustained efforts (including financial ones) are being made in order to put in practice the concerns related to valorisation of different by-products from the food industry, many things are still to be done to fulfil this desideratum. Grape pomace represents a valuable source of compounds that can be integrated by the food industry and pharmaceutics in different formulations. A first step in this demarche is a deep knowledge of the chemical composition of grape pomace that should be also, as much as possible, constant and/or easy to be brought to constant parameters. More than that, the producers do not have the time nor the often-needed infrastructure for the characterization of the grape pomace. For them, a certificate of conformity or a similar document could solve the problem of confidence in the quality of the raw material. In this sense, the present study aims to provide to the industry a tool for the utilization of grape extracts. The scientific substantiation of the chemical composition of extracts obtained from skin and seeds respectively were obtained by applying the vibrational spectroscopy analysis combined with multivariate analysis. The seed extracts of the four red grape varieties proved to be sources rich in phenolics and flavonoids, with a high antioxidant activity, while the skin extracts of the organic varieties of grape could be also considered for their bioactive compounds.

## Figures and Tables

**Figure 1 foods-10-01856-f001:**
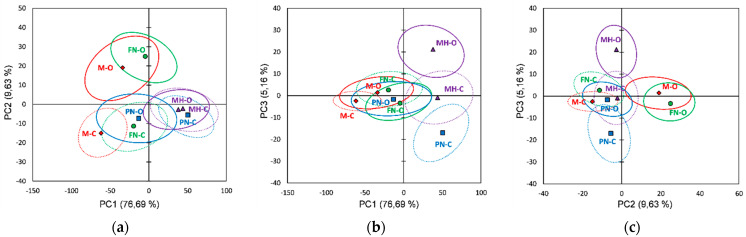
Score plots of the first three principal components derived from FTIR data of the red grape skin extracts (the confidence ellipses are based and centered on bootstrap points): (**a**) PC1 vs. PC2 score plot; (**b**) PC1 vs. PC3 score plot; (**c**) PC2 vs. PC3 score plot.

**Figure 2 foods-10-01856-f002:**
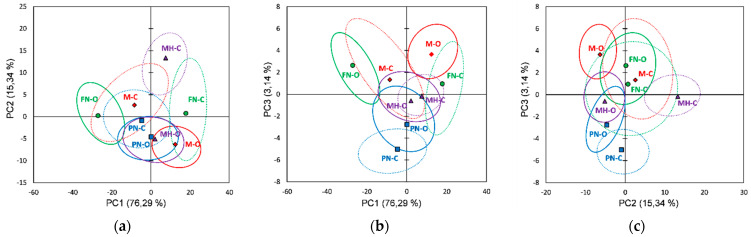
Score plots of the first three principal components derived from Raman data of the red grape skin extracts (the confidence ellipses are based and centered on bootstrap points): (**a**) PC1 vs. PC2 score plot; (**b**) PC1 vs. PC3 score plot; (**c**) PC2 vs. PC3 score plot.

**Figure 3 foods-10-01856-f003:**
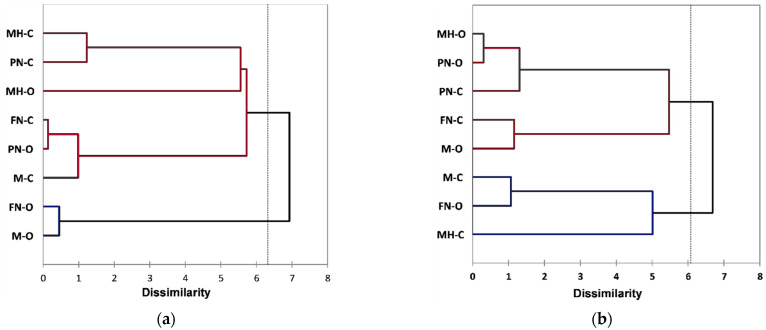
Dendrograms showing the cluster patterns for red grape skin extracts based on: (**a**) FTIR data; (**b**) Raman data.

**Figure 4 foods-10-01856-f004:**
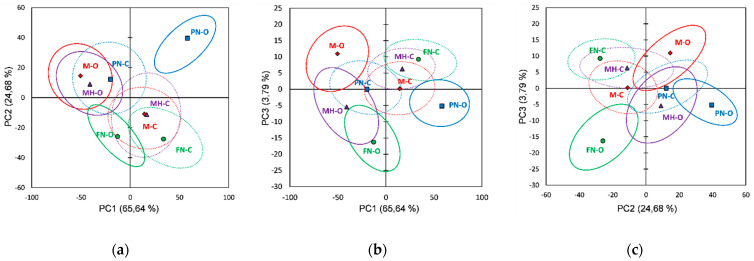
Score plots of the first three principal components derived from FTIR data of the red grape seeds extracts (the confidence ellipses are based and centered on bootstrap points): (**a**) PC1 vs. PC2 score plot; (**b**) PC1 vs. PC3 score plot; (**c**) PC2 vs. PC3 score plot.

**Figure 5 foods-10-01856-f005:**
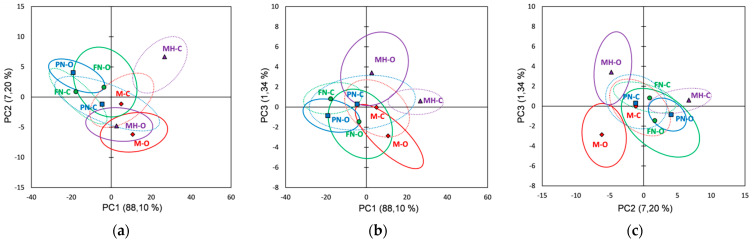
Score plots of the first three principal components derived from Raman data of the red grape seed extracts (the confidence ellipses are based and centered on bootstrap points): (**a**) PC1 vs. PC2 score plot; (**b**) PC1 vs. PC3 score plot; (**c**) PC2 vs. PC3 score plot.

**Figure 6 foods-10-01856-f006:**
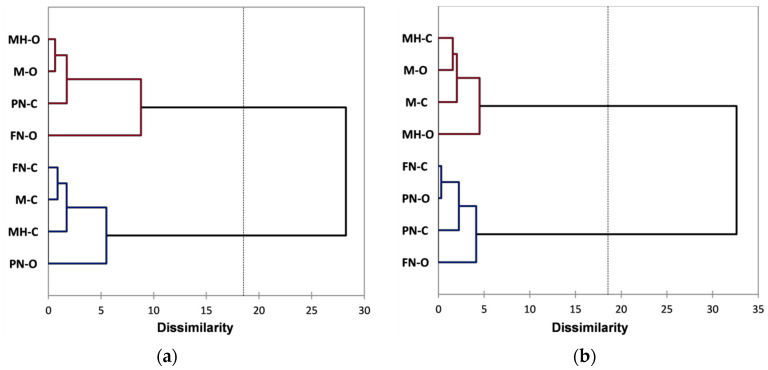
Dendrograms showing the cluster patterns for red grape seed extracts based on: (**a**) FTIR data; (**b**) Raman data.

**Figure 7 foods-10-01856-f007:**
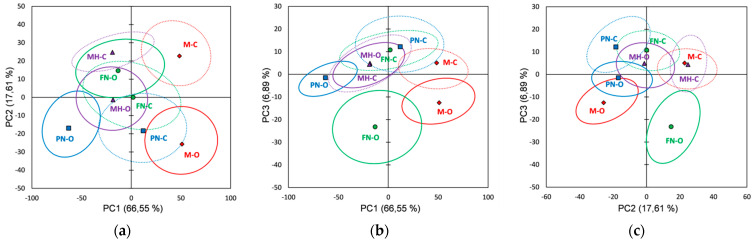
Score plots of the first three principal components derived from FTIR data of the red grape pulp extracts (the confidence ellipses are based and centered on bootstrap points): (**a**) PC1 vs. PC2 score plot; (**b**) PC1 vs. PC3 score plot; (**c**) PC2 vs. PC3 score plot.

**Figure 8 foods-10-01856-f008:**
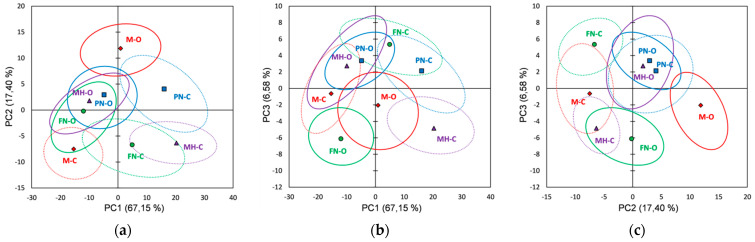
Score plots of the first three principal components derived from Raman data of the red grape pulp extracts (the confidence ellipses are based and centered on bootstrap points): (**a**) PC1 vs. PC2 score plot; (**b**) PC1 vs. PC3 score plot; (**c**) PC2 vs. PC3 score plot.

**Figure 9 foods-10-01856-f009:**
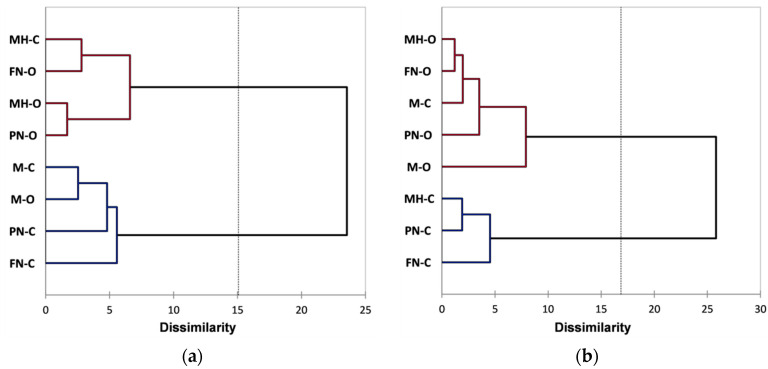
Dendrograms showing the cluster patterns for red grape pulp extracts based on: (**a**) FTIR data; (**b**) Raman data.

**Table 1 foods-10-01856-t001:** The red grapes (*Vitis vinifera* L.) investigated extracts.

Grape Variety	Vineyard Type	Sample Code
Merlot	Organic	M-O
	Conventional	M-C
Feteasca Neagra	Organic	FN-O
	Conventional	FN-C
Pinot Noir	Organic	PN-O
	Conventional	PN-C
Muscat Hamburg	Organic	MH-O
	Conventional	MH-C

**Table 2 foods-10-01856-t002:** Variability explained by the principal components (PCs) obtained by decomposition of the spectral data (red grape skin extracts) using principal component analysis (PCA).

PC Number	Variability [%]	
	FTIR Data	Raman Data
PC1	76.69	76.29
PC2	9.63	15.34
PC3	5.16	3.14
PC4	2.67	2.22
PC5	2.19	1.31
PC6	1.93	0.92
PC7	1.73	0.78

**Table 3 foods-10-01856-t003:** Prior and posterior classification of the red grape skin extracts using PC-DA (FTIR spectral data).

			Membership Probabilities	
Extract	Prior	Posterior	Pr (Conventional)	Pr (Organic)
M-O	Organic	Organic	0.000	1.000
FN-O	Organic	Organic	0.000	1.000
PN-O	Organic	Conventional	0.514	0.486
MH-O	Organic	Organic	0.000	1.000
M-C	Conventional	Conventional	0.999	0.001
FN-C	Conventional	Conventional	0.907	0.093
PN-C	Conventional	Conventional	1.000	0.000
MH-C	Conventional	Conventional	0.999	0.001

**Table 4 foods-10-01856-t004:** Prior and posterior classification of the red grape skin extracts using PC-DA (Raman spectral data).

			Membership Probabilities	
Extract	Prior	Posterior	Pr (Conventional)	Pr (Organic)
M-O	Organic	Organic	0.003	0.997
FN-O	Organic	Organic	0.000	1.000
PN-O	Organic	Organic	0.038	0.962
MH-O	Organic	Organic	0.031	0.969
M-C	Conventional	Conventional	1.000	0.000
FN-C	Conventional	Conventional	1.000	0.000
PN-C	Conventional	Conventional	1.000	0.000
MH-C	Conventional	Conventional	1.000	0.000

**Table 5 foods-10-01856-t005:** Variability explained by the principal components (PCs) obtained by decomposition of the spectral data (red grape seeds extracts) using principal component analysis (PCA).

PC Number	Variability [%]	
	FTIR Data	Raman Data
PC1	65.64	88.10
PC2	24.68	7.20
PC3	3.79	1.34
PC4	1.86	1.23
PC5	1.49	0.79
PC6	1.41	0.75
PC7	1.11	0.57

**Table 6 foods-10-01856-t006:** Prior and posterior classification of the red grape seeds extracts using PC-DA (FTIR spectral data).

			Membership Probabilities	
Extract	Prior	Posterior	Pr (Conventional)	Pr (Organic)
M-O	Organic	Organic	0.000	1.000
FN-O	Organic	Organic	0.000	1.000
PN-O	Organic	Organic	0.000	1.000
MH-O	Organic	Organic	0.000	1.000
M-C	Conventional	Conventional	1.000	0.000
FN-C	Conventional	Conventional	1.000	0.000
PN-C	Conventional	Conventional	0.949	0.051
MH-C	Conventional	Conventional	1.000	0.000

**Table 7 foods-10-01856-t007:** Prior and posterior classification of the red grape seeds extracts using PC-DA (Raman spectral data).

			Membership Probabilities	
Extract	Prior	Posterior	Pr (Conventional)	Pr (Organic)
M-O	Organic	Organic	0.000	1.000
FN-O	Organic	Organic	0.000	1.000
PN-O	Organic	Organic	0.000	1.000
MH-O	Organic	Organic	0.000	1.000
M-C	Conventional	Conventional	0.989	0.011
FN-C	Conventional	Conventional	0.994	0.006
PN-C	Conventional	Conventional	0.899	0.101
MH-C	Conventional	Conventional	1.000	0.000

**Table 8 foods-10-01856-t008:** Variability explained by the principal components (PCs) obtained by decomposition of the spectral data (red grape pulp extracts) using principal component analysis (PCA).

PC Number	Variability [%]	
	FTIR Data	Raman Data
PC1	66.55	67.15
PC2	17.61	17.40
PC3	6.89	6.58
PC4	2.98	4.14
PC5	2.44	2.66
PC6	2.09	1.12
PC7	1.42	0.93

**Table 9 foods-10-01856-t009:** Prior and posterior classification of the red grape pulp extracts using PC-DA (FTIR spectral data).

			Membership Probabilities	
Extract	Prior	Posterior	Pr (Conventional)	Pr (Organic)
M-O	Organic	Organic	0.000	1.000
FN-O	Organic	Organic	0.000	1.000
PN-O	Organic	Organic	0.000	1.000
MH-O	Organic	Organic	0.000	1.000
M-C	Conventional	Conventional	1.000	0.000
FN-C	Conventional	Conventional	0.991	0.009
PN-C	Conventional	Conventional	0.980	0.020
MH-C	Conventional	Conventional	0.998	0.002

**Table 10 foods-10-01856-t010:** Prior and posterior classification of the red grape pulp extracts using PC-DA (Raman spectral data).

			Membership Probabilities	
Extract	Prior	Posterior	Pr (Conventional)	Pr (Organic)
M-O	Organic	Organic	0.000	1.000
FN-O	Organic	Organic	0.003	0.997
PN-O	Organic	Organic	0.049	0.951
MH-O	Organic	Organic	0.040	0.960
M-C	Conventional	Conventional	0.722	0.278
FN-C	Conventional	Conventional	1.000	0.000
PN-C	Conventional	Conventional	1.000	0.000
MH-C	Conventional	Conventional	1.000	0.000

## Data Availability

The data presented in this study are available within the article or Appendix A.

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
