# Peer review of "Vibrational Spectroscopy Combined with Chemometrics as Tool for Discriminating Organic vs. Conventional Culture Systems for Red Grape Extracts"

_foods, 2021, doi:10.3390/foods10081856_

Round 1
Reviewer 1 Report
The paper describes a methodological work on the application of vibrational spectroscopies (IR and Raman) to the discrimination of different grape extracts.
The work is complete, the experimental plan seems sound and methods used are well described and adequate to the purpose presented.
I have some comments and I am suggesting some changes that in my opinion are needed in order to improve the significance and interest to readers.
Forst of all, several parts of the manuscript are unnecessarily lenghty and verbous. For example, in the Abstract I would delete lines 22-25, and 27-28. Abstract should concisely describe the results and data interpretation, not well known methods. The same can be said for Materials and Methods section, which is far too long so that the reader gets lost into the description (and discussion) on the statistical methods used. Lines 147-151 are not relevant in this context, moreover, they are difficult to understand and should be rephrased. I suggest to shorten especially the whole 2.3. Multivariate analysis part regarding statistical analysis of data. A more concise description of methods should be reported, since readers are usually trained to understand (or study) the differences between the methods used. I would just briefly focus on the reasons why such methods have been used. I'm not sure this part can be considered a part of Materials and methods section. I see it more suitable as a discussion.
The discussion should be thoroughly revised considering that what is really relevant to the reader is WHY the samples are grouped by following the methods presented. For example the authors linger too much on what spectral regions discriminate what groups of samples, but never explain what molecules/functional groups/key compounds/classes of compounds possess discriminant ability. And deepen the discussion on the reason why they lead to differentiate metabolic profiles of the different extracts.
I do not understand why the samples have been dried at high temperature (40°C) before extraction, and why the were macerated at room temperature for long time (22-23°C/24 hours), and why the extracts were stored at 5°C. All these processes contribute to degradation of key compounds. What would happen if the samples were treated in a proper sample preparation conditions so to avoid their oxidation? Is the clusterization of samples influenced by sample treatment?? Is the sample preparation method used a conventional protocol taken from industrial processes?
There are several spelling errors, grammatical errors and some phrases unintelligible or difficult to read. The manuscript should be checked very carefully.
Conclusions are quite superficial. I would like to know if this method provides useful information for improving the utilization of grape extracts by industry, and if yes, what differences are observed in key compounds among the different extracts.
Author Response
Please see the attachement

Reviewer 2 Report
This manuscript tried to discriminate the organic and conventional culture systems for red grape extracts from different red grapes berry parts by using FTIR and Raman combined with chemometrics analysis. Generally, there are too many information throughout the manuscript and some of the experimental designs are very confusing. It needs substantial restructure. My detailed comments are shown as follows:
Introduction: the introduction part should be more concentrated on the main focus on your study. For example, From Lines 50-81, I am not sure what the main point do the authors would like to express and what the relationship between your current research study and your objectives. As there are so many literatures have been studied the extracts from grape seed by using ATR-FTIR and Raman spectroscopy (such as: "Study of phenolic extractability in grape seeds by means of ATR-FTIR and Raman Spectroscopy"), what the novelty of your research study should be clearly clarified.
Material and Methods
This part is quite confusing and not well described. There are so many factors: 1) different anatomical parts (skin, seeds, pulp),
2) different cultivation systems (organic and conventional)
3) different grape varieties.
I am wondering how the authors avoid the interactive effects between those factors. Most importantly, the title is "discriminate organic vs. conventional culture systems", which means all other factors should be fixed and the independent variable is culture systems. Why were there so many factors to study?
The authors stated the reason for selecting the grapes, which made the readers more confusing: from Lines 152- 155, it referred to the climate, the hilly area, the altitudes, and the soil type, how many factors did you study? If you just want to state those factors will affect the growth of grapes, it should be put into the introduction part, but I believe it is not helpful to clarify what the main point of your study.
Line 202: what do you mean "without previous treatment"?
Results: I suggest the authors combine the results and discussion together to make the reader clearly understand. For example, from the Lines 335-359, it seems the discussion. Therefore, I strongly recommend the authors to combine them together, rendering it clearly to understand.
Line 321: There are no Figure S1 - S6 although it provides the link in Line 829, it cannot open. There are also so many repeatable information from tables and figures: such as Figure 1 and Table 2; Table 5 and Figure 4 and Table 8 and Figure 7 and so on. The most important the raw spectra from ATR-FTIR are not shown. Please restructure them. Also, the table should be three-line table.
The conclusion should provide the main and important finding from you study, better to present in numerical.
Reviewer 3 Report
This report from Radulescu C. et al. presents an interesting application of multivariate explorative analysis and discriminant models, in order to distinguish grape’s extracts from organic and conventional cultivation systems. The study is based on vibrational spectroscopies (i.e., FTIR and Raman), with a detailed interpretation of the principal vibrational modes characterizing the extracts.
Although the main idea has merit, and the obtained results conform to a reasonable application of the two techniques, several points must be addressed, both in the practical display of the results and in the overall structure of the study. Moreover, if part of the data had been already presented in previous similar publications, this should be stated clearly in the Introduction outlining the main past findings.
A list of revisions is provided here below, organized by sections. Considering these comments and observations, I would propose that major revisions are made to the manuscript before re-considering it.
LIST OF REVISIONS
- Abstract
- Check punctuation (e.g., “:..FTIR, and Raman screening…”).
- Lines 25. Here, “pattern” is used to indicate a statistical workflow, whereas it was used to mean “spectral features” one line before. This creates ambiguity.
- Line 26. “… Vibrational spectroscopic techniques, attenuated total reflectance-Fourier transform infrared (ATR-FTIR) and Raman…”. Here, one understands that three techniques were used.
- Line 30 and 31. “A well-defined differentiation based on red grape variety could not be highlighted.”. But this was not the aim of the work. According to the Discussion, varietal patterns could be observed (see Lines 741-776) in the spectral features. Did the Authors attempt to differentiate by variety? Otherwise, this comment in the Abstract would be pointless.
- Introduction
- The Introduction should cite and comment on related publications, also from the Authors themselves, especially if very similar in goals and methods as the present one. Besides, relevant literature on the use of FTIR and Raman scattering for grape authenticity assessment could be applied. The author should also comment on the advantages of analyzing the extracts instead of grape parts.
- Material and Methods
- Line 146-159. This is an introduction to the methods and study design; however, it is a really a kind of discussion. Please, pass discussions on the methods or material to the Discussion (leaving in M&M Table 1, as part as 2.1 paragraph).
- Could it be possible including some information on the grape clones, the level of irrigation (e.g. average mm/day of rain, irrigation...), level of light exposure of the two vineyards?
- Could it be possible including also some information on the treatments applied on the two vineyards?
- Line 150. “Both vineyards”: location of the vineyards have not been indicated previously in the report. Please, indicate the positions of the vineyards (maybe by GPS coordinates).
- Line 153. “hilly area”. Same comment as n.9.
- Line 156. Table 1 does not report phytochemical spectra profiles.
- Line 161 (and everywhere else). Were replicates included in the study?
- Paragraph 2.1: Please indicate:
1) Quantity of grape harvested per sample;
2) Was sampling representative of the entire vineyard? How the variability of the vineyard was ensured in the study? mixing grapes from different location in the vineyard, or replicates?
- Line 166. “Raw products of same batch were used, skin, seeds and pulp being manually separated from each sampled grape variety and vineyard”. What does it mean?
- Line 166: “Berry parts were first dried at 40ºC for 48 hours and 166 then stored at room temperature, in closed containers, protected from moisture and light, 167 for further experiments.” Is this treatment (in air, under heating) safe for the phenolics or the components in general? Can the Authors show that this treatment has no effects, or at least it does not influence the spectral features in such a way to influence the results of the multivariate analysis? It may also be that the different compositions (e.g. different phenolic content) of the different varieties could influence the outcomes of this treatment. Besides, how were these conditions optimized? Is that presented in another report?
- The entire paragraph focuses on the berry pulp. The Authors should explicitly say if skins and seeds were extracted in the same way (or else).
- Line 206. Some multivariate models are used to quantitate, not certainly the ones applied here.
- Line 208. Agreed. However, matrix interference effects are still detrimental, especially when the sample size is not large enough to properly average out his contribution (by creating systematic errors).
- Line 219. “The ATR correction algorithm can ‘symmetries’ the peak shape and when…”. Please, check English.
- Line 246. Please, explain what “IR synchrotron” means in this context or add a reference.
- Line 248. “Box-Cox transformation…”. Please, report explicitly the function applied to the data.
- Line 248-250: Throughout this section, it is not clear whether the transformation was applied to the spectra (along rows) or to each wavelength (along columns). I am aware that in some vibrational spectroscopies (e.g., NIR), SNV transformations along samples (raws) can be applied, however here I have the impression that the transformations were only applied along columns (Wavenembers or Raman shifts). Was I right?
- Line 258-259: What does “Based on results of the Principal Component Analysis (PCA) plots” mean?
- Line 307: “…and output tests on the variables assume 307 them to be normally distributed”. Please, check English.
- Results
GENERAL COMMENTS
Besides the following specific points, I would suggest to include here also the Loadings plots for the N components retained in the PCA-DA model (as bar plots), now placed in SI ,so to aid the understanding of which spectral features did play in making the PCA-DA models effective.
- The absence of a Figure including one FTIR and one Raman model spectrum in this paragraph renders it rather difficult to go through, especially for any reader with little or no expertise in infrared spectroscopy. I would propose that the Authors simply transfer here one or two example spectra, labelling the peaks of interest, and summarizing all the paragraph in one table, whereby including their interpretations on the bands in terms of vibrational modes. This would ease the reading of this section quite a lot.
- Line 335-336.Did not you use water/ethanol for the extraction? Did that not contribute?
- Line 339. “…was actually recorded”. What does it mean? Please, rephrase.
- Line 339-340. “The spectral band located at 2979 cm-1 can be associated with the solvent 339 (ethanol) being due to the stretching vibrations of the O-H groups”. I could be wrong, but I fear this band should be attributed to C-H stretching instead.
- Line 343-344. “the presence of this maxim”. What is a “maxim”?
30. Line 346. Why just polysaccharides? If you are now including also these compounds among the possible ones extracted with your method, can you be so specific in assigning this band? - Line 365-367. Could not also other compounds contribute for seeds? Are not also lipids extracted with the used method from seeds?
- Table 2. Please, show just Variability or Cumulative variability. Showing both is just redundancy.
- Line 394-403. How it is possible to have confidence ellipses (created in any possible way) distributions and statistical hypotheses built over only one not-replicated observation per group (not even technical replicates or maybe samples from the same location, not to say different locations), especially when they encompass just one sample?!?! It does not make much sense, if not for a qualitative/interpretative purpose.
Besides, of course two individual observations are different… they are two different objects; indeed, how could they be identical? From this, to say that they belong to different distributions is rather another story and should not be considered anywhere here for the interpretation of the results.
- “The software (XLSTAT) calculates and represents for each original observation the 95% confidence ellipse based and centered on the bootstrap points”. “It can be concluded that two observations are significantly different from each other on a given PC plan if their ellipses do not overlap”. In all these instances, see the previous comment.
- Figure 1. I think the use of the ellipses is really superfluous! Please, remove them.
- From Line 407. Please, rewrite this section excluding all reference to the ellipses.
- For all PCA. No trends are evident in any of the PCAs, still for the aforementioned points (see 33) – no replicates. Besides, no neat organic/conventional or varietal separations can be observed. Hence, discussion about trends in PCA should be avoided everywhere.
- Line 465. Please, indicate if eventually LDA or QDA has been used, as within-variances were indeed different.
- Table 3. Various comments are in order. First: Was a variable selection applied? Secondly: Could you comment on the difference between FTIR and Raman in the model performance and outline the most relevant spectral features allowing the model to work in the two cases? To which vibrational modes/chemical components are these bands mostly belonging? Could then a variable selection improve the models performances even more?
- For all the remaining Results paragraph, the previous Results comments apply.
- Discussion
Please, reconsider the discussion paragraph after all these changes. Besides, please comment mostly on the common (between varieties) spectral features that differentiated for the organic/conventional growing factor. As the PCA, and therefore the DA, models where built including all samples (not by specific variety), one could imagine these features differentiating by cultivation mode should be the same regardless of the variety. Is that correct?
Besides, it would be useful commenting on which signals were the most useful in actually differentiating the organic vs. conventional samples, as you applied PCA and PCA-DA models on all varieties together. Maybe also comparisons between the Raman and FTIR in terms of relevant vibrational modes for the discrimination.
Finally, the discussion paragraph should be simplified. Direct descriptions of the spectra/plots might be moved in the result section.
Round 2
Reviewer 1 Report
The manuscript has been somewhat improved by the revision. However, still some relevant criticisms have to be highlighted.
Many errors are still present. Just to make some examples (referring to line numbers within the MS with tracked changes):
Line 265: can be more accurately investigated by multivariate analysis.
Line 266: Some multivariate models provide a means
Line 453: ATR correction algorithm can ‘symmetrize’
Line 1033: deep knowledge
Lines 445-546 can be moved to Supplementary information. It has nothing to do with experimental results, but describe procedures (methods) used. This part is very clear and well described but, you must agree, it does not describe results obtained on the systems under study. The same can be said for Lines 561-572. In this latter case, I would not move the whole part, but it can be significantly simplified in the text, and all methodological details can be moved to Supplementary information. Please also do the same (simplify the text) at lines 625-645.
The following paragraphs have the same structure and appear to be very monotonous (Lines 764-773; 742-757; 686-701; 665-679; 600-612; 579-591);. They explain how the different clusters of samples are separated or superimposed in scores plots. These parts should be condensed significantly, please mention (or briefly explain/comment) just the most relevant data. It is quite clear from the observation of Figures how the different clusters separate in the graphs. Some parts from Discussion section should be moved here (see below).
The first sentence in Discussion (Lines 802-812) is way too long. Please consider to divide it into two shorter sentences. Also its lexicon should be revised.
Lines 829-869: This part is not a discussion of obtained data, but represents the data themselves (they are not raw data, but processed data). Therefore, it should be moved to Results section. The same can be said for the larger part of Discussion. I hardly see any relevant (and interesting) discussion of the (very interesting!) data acquired.
Some sentences need further comment. Just to make an example, the sentence "Excepting Pinot Noir, the rest of red grape varieties shows notable difference between 870 organic and conventional vineyards (M-O vs. M-C, FN-O vs. FN-C and respectively 871 MH-O vs. MH-C), the corresponding extracts for each variety, organic and conventional, 872 being assigned in the two main clusters." should be commented. Do the authors have any clue to explain this observed result? It would be very interesting to investigate on the reason why Pinot Noir shows different behaviour from other cultivars. By the way, please revise grammatical errors (e.g. in the abovementioned sentence: "the rest of red grape varieties show") and lexicon. For example, I would rewrite the same sentence as: "Except Pinot Noir, red grape varieties show notable differences between organic and conventional vineyards." No more detail is needed in the Discussion, since all samples numerosity, data, Tables and Figures have been reported in the Results section.
Reviewer 2 Report
As I mentioned before, the introduction part should be more concentrated on the main focus on your study. In the revised version, the authors don't have any improvement. Also there are repeatable information such as Table 5 and Figure 4, Table 8 and Figure 7 and so on. As long as you have so many figures which also put in the Supplementary Materials, why don't you restructure them and delete the repeatable information and show the most important information? For the conclusion, it is still like the discussion part, especially for the adding sentence. As aforementioned, it should be concise and put the most important information but the authors failed to address my comments. For the table, it should be used the three-line table but the authors still do not improve it. The next when you reply to the referee, please also mark where it is in the revised manuscript, which will save the time for the referee to find the answers.
Reviewer 3 Report
The changes made by the Authors improved the manuscript. However, as the Authors stated that replicates of each extract were taken, I strongly suggest to include all these replicates in the all the PCA as observations, and not to consider their averages/means: using averages could really be misleading and give rise to false interpretations of the results: as mentioned, testing two single objects for significant difference is not as comparing two groups/distributions - in the former case, the information about the standard deviations within each group is completely lost - and of course only the second case should be accepted. It is also not particularly useful here, as the number of samples (even considering the replicates) is surely not too big. Eventually, confidence ellipses could be created by another convenient option available for PCA in XLStat (Charts - Observations), which may really account for the variability within each group of replicates (so representing true variability withing and between groups of replicates -albeit just technical ones- and not a statistically reconstructed one, as with the bootstrapping approach). This might way represent real information and valuable interpretation.
